# FLOW-IB: INFORMATION BOTTLENECK MEETS FLOW MATCHING FOR 32,768× VIDEO COMPRESSION

## ABSTRACT

We present a generative video compression framework that achieves an unprecedented 32,768× compression ratio by transmitting only the first and last frames as I-frames and reconstructing the remaining content with a flow-matching video diffusion model. Guided by the information bottleneck principle, our method introduces a differentiable loss that minimizes redundant information with the known I-frames, enabling joint optimization of compression and generation within a unified framework. This design allows the generative model to faithfully reconstruct intermediate frames at extreme compression rates. Extensive experiments demonstrate that our approach substantially outperforms both traditional codecs and recent deep learning–based schemes across standard rate–distortion metrics. Moreover, the reconstructed videos deliver comparable performance to state-of-the-art semantic communication methods across multiple downstream tasks, demonstrating the strong potential of generative compression as a practical alternative to conventional coding.

## 1 INTRODUCTION

The exponential growth of video data, fueled by high-resolution capture devices and ubiquitous video-based applications, has placed unprecedented demands on storage and transmission infrastructures. Current video compression standards, including H.266/VVC (Zhang et al., 2020) and AV1 (Chen et al., 2020), have achieved substantial improvements through decades of engineering refinement, employing hybrid coding strategies that combine motion estimation, transform coding, and entropy modeling. However, at extremely low bitrates, these approaches inevitably encounter a fundamental cliff in performance, manifesting as severe blocking artifacts, blurriness, and a catastrophic loss of high-frequency details. This precipitates a paradigm shift from purely fidelity-based reconstruction towards perceptual reconstruction, where the goal is to preserve the semantic integrity and visual plausibility of content for human perception, even if pixel-level accuracy is sacrificed.

Currently, generative models have demonstrated remarkable capabilities in video synthesis and reconstruction. These models are increasingly being leveraged to achieve faithful content reconstruction under severe bitrate constraints by capitalizing on their powerful inherent priors in conjunction with compact spatio-temporal guidance. Existing methods (Zhang et al., 2025; Wan et al., 2024; Wu et al., 2023; Yi et al., 2025) often rely on hand-crafted features, such as sketches, canny edges, or textual prompts, to represent the video content. However, these low-level, static descriptors are insufficient to capture the rich dynamics and high-level semantics of video content, inevitably leading to information loss. Moreover, the decoupled optimization of the feature extractor and the reconstruction model leads to a suboptimal system. The extraction process is agnostic to the final reconstruction goal, resulting in a mismatch between the extracted features and the information most critical for the generative model. Consequently, reconstructed videos often lack fidelity, particularly in dynamic and complex scenes.

In this work, we propose a trainable video compression and transmission framework, in which the feature extraction network at the transmitter side and the generative network at the receiver side are jointly optimized in an end-to-end manner. This design allows the feature extractor to leverage the strong prior of the generative model, thus explicitly capturing the information most essential for the faithful video reconstruction. To leverage the strong temporal coherence inherent in video data, we treat the first and last frames of each video as known information. These frames, often referred to

as I-frames in conventional video coding, are transmitted to the receiver. They provide essential conditioning signals to the generative model, facilitating high-fidelity reconstruction of the entire video sequence.

Our framework is motivated by the information bottleneck principle (Tishby et al., 2000; Tishby & Zaslavsky, 2015), as it provides a theoretically grounded insight for optimizing the trade-off between compression and reconstruction, and has been widely applied across various representation learning tasks (Absar Siddiky et al., 2024; Shao et al., 2021). Starting from its original formulation, we derive a tractable variational bound as our optimization objective. Furthermore, we introduce a reformulated flow-matching objective that unifies the training of the generative model and the ultimate reconstruction task. Ultimately, our approach yields a minimally sufficient representation of the video, achieving an extreme compression ratio of 32,768× compared to the original input.

To summarize, our contributions are threefold:

- We introduce the first end-to-end video compression framework that integrates generative models with information bottleneck principles, enabling joint optimization of compression and reconstruction quality.
- We reformulate the flow-matching objective into a unified training paradigm that simultaneously optimizes the generative model and its feature extractor, leading to more coherent and efficient representations.
- Our method achieves an unprecedented compression ratio of ×32,768, while consistently outperforming existing approaches across diverse downstream applications.

## 2 RELATED WORK

**Video Compression.** Video compression aims to reduce redundant information while preserving critical visual content, enabling efficient storage and transmission for applications such as streaming and video conferencing. With the success of deep learning in image compression (Mishra et al., 2022), neural video compression methods have emerged to optimize rate-distortion performance. For example, residual coding approaches (Choi & Bajić, 2019) generate predictions from previously decoded frames and encode the residuals. Subsequent end-to-end learned frameworks like the DCVC series (Li et al., 2021; 2023; 2024) further improve compression performance and outperform traditional codecs such as ECM (Karadimitriou, 1996). However, these methods primarily focus on pixel-level distortion metrics and often overlook perceptual quality, particularly at extremely low bitrates, where they encounter a severe performance cliff. This limitation has motivated research on the rate-distortion-perception trade-off (Blau & Michaeli, 2019), spurring research into perceptual video compression.

**Generative Models for Video Compression.** Recent advances in generative image compression have inspired growing interest in their application to video. Pioneering work includes that of Wu et al. (2023), who transmitted sketches and text descriptions to guide diffusion-based reconstruction. Careil et al. (2023), who used vector-quantized latents and captions for decoding. Yi et al. (2025) proposed a multi-control framework that compresses segmentation, optical flow, and pose into a compact representation and jointly drives video generation. To improve efficiency, Relic et al. (2025) formulated quantization noise removal as a denoising task with adaptive steps. This body of work highlights the potential of conditional generative modeling for compression, yet existing methods often rely on hand-crafted cues (e.g., sketches or text) and lack end-to-end optimization. Our method addresses these gaps by introducing a learned conditional latent prior and a fully differentiable compression pipeline, enabling efficient high-fidelity reconstruction at extreme ratios.

## 3 METHOD

### 3.1 PRELIMINARY

The goal of video compression is to minimize the number of bits required to represent a video sequence while maximizing the perceptual quality of its reconstruction. Let $\mathbf{X} \in \mathbb{R}^{T \times H \times W \times 3}$ denote a source video sequence of $T$ frames, and $\mathbf{C} \in \mathbb{R}^{T' \times H \times W \times 3}$ represent the conditioning information, which in our case is a subset of frames from $\mathbf{X}$ (e.g., the first and last frames). The objective

Figure 1: An overview of the proposed generative video compression framework. (a) Training: The source video is encoded into latent representation $\mathcal{X}$ by a pretrained VAE Wan (2025), then processed by a U-Net styled compressor (with down and up modules) that outputs Gaussian parameters. These are used to: (1) compute KL loss against a temporally masked prior. (2) combined with conditioning frames to reconstruct the latent representation via a generative model after reparameterization. (b) Inference: On the transmitter side, the video is further compressed into a transmitted representation $h$ by the compressor's down module to reduce bandwidth. The first/last frames and $h$ are then sent via the network channel. On the receiver side, $h$ will be processed with compressor's up module and reconstructs the video using the generative model and VAE decoder.

is to learn a minimal but sufficient encoding $\mathbf{Z}$ derived from $\mathbf{X}$, such that when combined with $\mathbf{C}$, it enables a generative model to generate a high-quality reconstruction. The intuition is that $\mathbf{Z}$ should discard all redundant information with $\mathbf{C}$, preserving only the novel, dynamic content necessary for faithful reconstruction. This goal can be formally expressed by minimizing the following objective according to the conditional information bottleneck principle (Fischer, 2020):

$$\mathcal{L}_{\text{CIB}} = \underbrace{H(\mathbf{X} \mid \mathbf{Z}, \mathbf{C})}_{\text{Reconstruction Term}} + \underbrace{\beta\, I(\mathbf{Z}; \mathbf{X} \mid \mathbf{C})}_{\text{Compression Term}}, \tag{1}$$

where $H(\mathbf{X}|\mathbf{Z}, \mathbf{C})$ is the conditional entropy, representing the uncertainty in reconstructing $\mathbf{X}$ given $\mathbf{Z}$ and $\mathbf{C}$. Minimizing this term encourages high fidelity. $I(\mathbf{Z}; \mathbf{X}|\mathbf{C})$ is the conditional mutual information, measuring the information shared between $\mathbf{Z}$ and $\mathbf{X}$ given $\mathbf{C}$. Minimizing this term encourages compression. The hyperparameter $\beta > 0$ controls the trade-off between reconstruction fidelity and compression.

In the following subsections, we introduce a tractable approximation derived from this objective (Sec. 3.2), detail the architectural instantiation of each component (Sec. 3.3), and describe the end-to-end training procedure (Sec. 3.4).

## 3.2 INFORMATION-THEORETIC FOUNDATION

The optimization objective of Eq. (1) is intractable to optimize directly. We therefore derive an approximation to enable efficient optimization.

We begin with the definition of the conditional entropy $H(\mathbf{X}|\mathbf{Z}, \mathbf{C})$:

$$H(\mathbf{X}|\mathbf{Z}, \mathbf{C}) = -\mathbb{E}_{p(\mathbf{x},\mathbf{z},\mathbf{c})}\left[\log p(\mathbf{x}|\mathbf{z}, \mathbf{c})\right] \quad \text{(definition of conditional entropy)} \tag{2}$$

$$\leq -\mathbb{E}_{p(\mathbf{x},\mathbf{z},\mathbf{c})}\left[\log p_\theta(\mathbf{x}|\mathbf{z}, \mathbf{c})\right] \quad \text{(variational approximation)} \tag{3}$$

Here, $p_\theta(\mathbf{x} \mid \mathbf{z}, \mathbf{c})$ is a flexible approximate variational distribution with parameters $\theta$ that we seek to optimize. Intuitively, it can be thought of as the generative model that reconstructs a high-quality video from the compressed code $\mathbf{z}$ and the conditioning information $\mathbf{c}$.

For the conditional mutual information $I(\mathbf{Z}; \mathbf{X}|\mathbf{C})$:

$$I(\mathbf{Z}; \mathbf{X}|\mathbf{C}) = \mathbb{E}_{p(\mathbf{x},\mathbf{c})}\left[D_{\text{KL}}\left(p(\mathbf{z}|\mathbf{x}, \mathbf{c}) \,\|\, p(\mathbf{z}|\mathbf{c})\right)\right] \quad \text{(definition of mutual information)} \tag{4}$$

$$= \mathbb{E}_{p(\mathbf{x},\mathbf{c})}\left[D_{\text{KL}}\left(p(\mathbf{z}|\mathbf{x}) \,\|\, p(\mathbf{z}|\mathbf{c})\right)\right] \quad \text{(c is a subset frames of x, negligible)} \tag{5}$$

$$\leq \mathbb{E}_{p(\mathbf{x},\mathbf{c})}\left[D_{\text{KL}}\left(q_\phi(\mathbf{z}|\mathbf{x}) \,\|\, p(\mathbf{z}|\mathbf{c})\right)\right] \quad \text{(variational approximation)} \tag{6}$$

Here, $q_\phi(\mathbf{z}|\mathbf{x})$ can be thought of as a learnable compressor network with parameters $\phi$ that is designed to map the full input video $x$ to the compressed code $z$. $p(\mathbf{z}|\mathbf{c})$ is a prior distribution, it characterizes what $z$ is expected to look like when $c$ is given.

Substituting Eq. (3) and Eq. (6) yields our tractable loss:

$$\mathcal{L} = \underbrace{-\mathbb{E}\left[\log p_\theta(\mathbf{x}|\mathbf{z}, \mathbf{c})\right]}_{\text{Reconstruction Term}} + \beta \underbrace{\mathbb{E}\left[D_{\text{KL}}\left(q_\phi(\mathbf{z}|\mathbf{x}) \parallel p(\mathbf{z}|\mathbf{c})\right)\right]}_{\text{Compression Term}} \tag{7}$$

### 3.3 Model Architecture and Implementation

In this section, we will describe the architectural instantiation of the prior distribution $p(z|c)$, the compressor $q_\phi$, and the generative model $p_\theta$, respecting Eq. (7).

**Prior distribution** $p(z|c)$**.** A key innovation of our work lies in the design of the prior distribution, which aims to guide the compressor to discard any information already available in the conditioning frames $\mathbf{c}$ (i.e., the first and last frames). This ensures that the compressed code $\mathbf{z}$ captures only the novel content present in the intermediate frames.

To achieve this, we introduce a simple yet effective temporal masking strategy. Our insight is that the essential dynamic information is primarily contained in the intermediate portions of the video. We therefore construct a masked version of the video $\tilde{\mathbf{x}}$, by setting the first and last frames to zero, preserving only the intermediate content. This masked video explicitly represents the necessary dynamic information, providing an appropriate prior for guiding the compression process.

To further help compute the KL divergence term, the masked video is then processed by the pre-trained VAE (Wan, 2025), which parameterizes it into a Gaussian distribution $p(\mathbf{z}|\mathbf{c}) = \mathcal{N}(\boldsymbol{\mu}_p, \boldsymbol{\sigma}_p^2\mathbf{I})$ with $\boldsymbol{\mu}_p, \boldsymbol{\sigma}_p = \text{VAE}(\tilde{\mathbf{x}})$.

**Compressor** $q_\phi$**.** For computational efficiency and motivated by the common practice of modern generative models operating in latent space, the compressor takes as input the latent representation $\mathcal{X}$ produced by the pretrained VAE (Wan, 2025), rather than the raw video. The compressor outputs the parameters $(\boldsymbol{\mu}_q, \boldsymbol{\sigma}_q)$ for the Gaussian posterior distribution $\mathbf{q}$. A compressed code $\mathbf{z}$ can be sampled from $\mathbf{q}$.

Thus, the KL divergence between $q_\phi(\mathbf{z}|\mathbf{x})$ and $p(z|c)$ has a closed-form solution, leading to stable and efficient training:

$$\mathcal{L}_{\text{KL}}(q_\phi \parallel p) = \log\frac{\sigma_p}{\sigma_q} + \frac{\sigma_q^2 + (\mu_q - \mu_p)^2}{2\sigma_p^2} - \frac{1}{2}$$

**Generative model** $p_\theta$**.** To realize the mapping from the compressed code to the reconstructed video frames, we employ a generative model. While the choice of generative architecture is generally flexible, our approach utilizes the first-and-last-frame conditional model introduced by Wan (2025), as it inherently supports conditioning on the start and end frames. This design eliminates the need for additional conditioning mechanisms and simplifies the overall implementation. As a flow-matching-based generative model, it is designed to predict the velocity field that transports noise to the data manifold of $\mathbf{x}$. We reformulate the flow-matching objective to align it with the reconstruction goal. Instead of learning a path from noise to data, we learn a straight path from the compressed code $\mathbf{z}$ to the target data $\mathbf{x}$. We define the probability path as a linear interpolation:

$$x_t = (1 - t)x + t * z, \qquad t \in [0, 1].$$

The corresponding velocity field that generates this path is given by the time derivative:

$$\frac{d\mathbf{x}_t}{dt} = \mathbf{z} - \mathbf{x}.$$

Then, the generative model $v$ with parameters $\theta$ is trained to regress this vector field using a simple mean-squared error objective:

$$\mathcal{L}_{\text{G}} = \mathbb{E}_{t,\mathbf{x},\mathbf{z},\mathbf{c}} \left\| v_\theta(\mathbf{x}_t, t, \mathbf{c}) - (\mathbf{z} - \mathbf{x}) \right\|^2.$$

Notably, this reformed objective is mathematically equivalent to minimizing the MSE between $\mathbf{z}$ and $\mathbf{x}$, thereby establishing a direct link between flow-based generative modeling and the maximum-likelihood objective of the reconstruction term. Moreover, the linearity of the probability path from

z to x allows for high-fidelity reconstruction in very few sampling steps (e.g., fewer than 10), as validated empirically in Section 4.4. This offers a substantial advantage in inference efficiency compared to conventional noise-to-data sampling paradigms.

### 3.4 END-TO-END TRAINING AND INFERENCE

The complete pipeline of our proposed framework is illustrated in Figure 1. It should be noted that to achieve the extreme compression ratios required for ultra-low bitrate transmission, the compressor $q_\phi$ is deliberately designed using a U-Net architecture with a downsample module and an upsample module. The downsample module is responsible for producing the transmitted representation $h$ for network transmission. Correspondingly, the upsample module is utilized to reconstruct the spatial-temporal resolution $h$, ensuring compatibility with the subsequent processing stages. During inference, the downsampling module will be deployed on the *transmitter* side, while the upsampling part will be deployed on the *receiver* side. The detailed architecture of the compressor is illustrated in Figure 5 in Appendix.

The latent representation $\mathcal{X}$ produced by the pre-trained VAE has already been reduced by a factor of 4×8×8 in the original video dimensions. We apply an additional 2×8×8 spatio-temporal downsampling via the compressor's downsampling module, resulting in a total compression factor of 32,768× for the transmitted representation $h$ (calculated as $4 \times 8 \times 8 \times 2 \times 8 \times 8$). To ensure computation and transmission efficiency, $h$ has a dimension of 16 and is represented in the bfloat16 format, this leads to an ultra-low bitrate of 0.0078 bpp[1] for transmitting $h$.

During training, the compressed code $\mathbf{z}$ is sampled using the reparameterization trick to allow gradient backpropagation and introduce stochasticity. During inference, we use the mean $\boldsymbol{\mu}_q$ directly as $\mathbf{z}$, following common practice in prior work (Kingma & Welling, 2022).

$$z = \begin{cases} \mu_q + \boldsymbol{\sigma}_q * \mathcal{N}(0,1), & \text{training} \\ \mu_q & \text{inference} \end{cases} \quad (8)$$

For the generative model, we do not update all parameters, but instead employ Low-Rank Adaptation (LoRA) to efficiently fine-tune the pre-trained weights.

## 4 EXPERIMENTS

To comprehensively evaluate our approach, we first assess the visual quality of reconstructed videos on standard video compression benchmarks. Specifically, we report performance under low bitrate settings on established datasets including HEVC Class B, HEVC Class C (Boyce et al., 2010), UVG (Mercat et al., 2020), and MCL-JCV (Wang et al., 2016).

Furthermore, to evaluate semantic fidelity, that is, the preservation of high-level information crucial for machine perception, we tested the reconstructed videos on multiple downstream tasks. These include action recognition on Kinetics (Carreira & Zisserman, 2017), multiple object tracking (MOT) on MOT17 (Milan et al., 2016), and video object segmentation (VOS) on DAVIS2017 (Pont-Tuset et al., 2018). This demonstrates the practical utility of our compressed representations beyond pixel-level metrics.

To demonstrate the strong generalization capability of our approach, all experimental results were obtained using a **single, unified model without any dataset- or task-specific fine-tuning**. Its consistent performance across diverse tasks and datasets highlights the robustness of our method and underscores the promising potential of generative models in reconstruction-based compression. Specifically, we employ the open-source Koala (Wang et al., 2024) dataset, from which 50,000 clips are randomly sampled as our training set. We fine-tune the pre-trained Wan2.1-14B for 3 epochs with a learning rate of $2 \times 10^{-5}$ and a batch size of 8. We set rank ($r$) and alpha ($\alpha$) to 16 for LoRA. We set the hyper-parameter $\beta = 10^{-4}$. All videos were resized to 480p resolution; we segmented each input video into shorter clips of 45 frames, which were then compressed and reconstructed independently by our model.

---

[1] $\mathrm{bpp} = \frac{16\,(\text{dimension}) \times 2\,(\text{bytes for bfloat16}) \times 8\,(\text{bits per byte})}{32{,}768\,(\text{spatial-temporal compression})} \approx 0.0078$

## 4.1 EXPERIMENTAL SETTINGS ON VIDEO COMPRESSION BENCHMARK

**Evaluation Metrics.** For directly assessing the visual quality of the reconstructed videos, we use two commonly used metrics for evaluation: Fréchet Video Distance (**FVD**) and Learned Perceptual Image Patch Similarity (**LPIPS**), as they better align with human perception compared to traditional measures.

**Baseline Methods.** We compare the proposed framework with both traditional video compression standards, H.264, H.265, H.266, and SOTA video compression methods: (1) traditional video compression methods (H.264 (Wiegand et al., 2003), H.265 (Sullivan et al., 2012), H.266 (Bross et al., 2021)); (2) neural video compression methods (DCVC-DC (Li et al., 2023), DCVC-FM (Li et al., 2024), DCVC-RT (Jia et al., 2025)); (3) diffusion-based video compression methods (T-GVC (Wang et al., 2025), Multi-C (Yi et al., 2025)).

## 4.2 EXPERIMENTAL SETTINGS ON SEMANTIC COMMUNICATION BENCHMARK

**Evaluation Metrics.** For action recognition, we employ the Top-1 accuracy (P@1) as the evaluation metric. For multiple object tracking (MOT), we adopt widely-used metrics including MOTA (Multiple Object Tracking Accuracy) (Kasturi et al., 2009), MOTP (Multiple Object Tracking Precision), FN (False Negatives), and IDF1. For video object segmentation (VOS), we evaluate performance using the Jaccard index $\mathcal{J}$, contour accuracy $\mathcal{F}$, their average ($\mathcal{J}\&\mathcal{F}$), and contour recall ($\mathcal{F}$-Recall).

**Downstream Task Models.** For action recognition evaluation, we utilize TSM (Lin et al., 2019) as the downstream model. For multiple object tracking (MOT), we adopt ByteTrack (Zhang et al., 2022) for performance measurement. For video object segmentation (VOS), the evaluation is performed using XMem (Cheng & Schwing, 2022) along with its officially released pretrained weights.

**Baseline Methods.** We compare our method with traditional codecs, namely H.265/HEVC (Sullivan et al., 2012) and H.266/VVC (Bross et al., 2021), learnable codecs, namely FVC (Hu et al., 2021), PLVC (Yang et al., 2022) and DCVC-DC (Li et al., 2023), as well as recent semantics-oriented coding methods ROI (Cai et al., 2021), JPD-SE (Duan et al., 2022), SMC (Tian et al., 2023), and FreeVSC (Tian et al., 2024).

## 4.3 OVERALL PERFORMANCE

To accurately estimate the actual transmission bandwidth for fair comparison, the first and last frames were compressed using the state-of-the-art LIC (Li et al., 2025) method, resulting in a final bitrate of approximately 0.01 bits per pixel (bpp) for the entire compressed video representation. Note that after compression, the first and last frames occupy only 4 KB, contributing an additional 0.002 bpp ($4 \times 1024 \times 8 \div 45 \div 480 \div 720$). Please note that during inference, the steps for Multi-C are set to 50 while ours are 10.

**Visual Fidelity.** Table 1 summarizes the performance of our end-to-end framework alongside comparative methods across multiple datasets. Under equivalent bpp conditions, our approach consis-

Table 1: The overall performance of different methods on the test sets. Here, we report the FVD and LPIPS among actual videos and predictions generated by different methods. The best results are highlighted in **bold**, and the second best results are underlined. For the method Multi-C, we report its performance at a bpp of 0.0067, while all other methods are evaluated at a bpp of 0.01.

| Method | HEVC Class B | | HEVC Class C | | UVG | | MCL-JCV | |
|---|---|---|---|---|---|---|---|---|
| | FVD ($\downarrow$) | LPIPS ($\downarrow$) | FVD ($\downarrow$) | LPIPS ($\downarrow$) | FVD ($\downarrow$) | LPIPS ($\downarrow$) | FVD ($\downarrow$) | LPIPS ($\downarrow$) |
| H.264 (Wiegand et al., 2003) | 2738 | 0.8283 | 3022 | 0.7724 | 4252 | 0.8274 | 4844 | 0.7139 |
| H.265 (Sullivan et al., 2012) | 1327 | 0.4941 | 1452 | 0.4833 | 1688 | 0.4052 | 1030 | 0.3667 |
| H.266 (Bross et al., 2021) | 1022 | 0.4038 | 1273 | 0.4543 | 1052 | 0.2834 | 973 | 0.3217 |
| DCVC-DC (Li et al., 2023) | 892 | 0.3821 | 1135 | 0.4327 | 992 | 0.2742 | 887 | 0.3152 |
| DCVC-FM (Li et al., 2024) | 837 | 0.3672 | 1079 | 0.4186 | 968 | 0.2689 | 849 | 0.3103 |
| DCVC-RT (Jia et al., 2025) | 783 | 0.3543 | 1029 | 0.4058 | 947 | 0.2635 | 811 | 0.3068 |
| T-GVC (Wang et al., 2025) | - | 0.3512 | - | 0.3642 | - | 0.2212 | - | 0.3145 |
| Multi-C (Yi et al., 2025) | 597 | 0.2813 | 402 | 0.2625 | 571 | 0.2208 | **515** | 0.2926 |
| **Ours (bpp ≈ 0.01)** | **355** | **0.1932** | **311** | **0.1872** | **437** | **0.1645** | 561 | **0.1947** |

tently outperforms both traditional codecs (H.264, H.265, H.266) and recent neural compression baselines by a considerable margin. Notably, our method achieves state-of-the-art results on HEVC Class B, HEVC Class C, and UVG datasets across all reported metrics. On the MCL-JCV dataset, we attain the best LPIPS score while achieving competitive, second-best performance in FVD.

Although Multi-C (Yi et al., 2025) incorporates multiple hand-crafted features and delivers competitive results on conventional metrics, qualitative comparisons reveal advantages of our approach. As shown in Figure 2, our method more accurately reconstructs perceptually challenging elements, such as newly emerging objects, complex lighting effects, intricate textures, and subtle tonal variations. Such elements are inherently difficult to model using hand-crafted features. These results demonstrate the superiority of our data-driven, end-to-end learning framework.

**Semantic Fidelity.** Table 2 summarizes the performance on downstream tasks, including action recognition, multi-object tracking, and video object segmentation, using videos reconstructed by different compression methods. For the action recognition task, our method consistently outperforms all competing approaches on the large-scale Kinetics dataset when using the TSM model, despite operating at a significantly lower bitrate (0.01 bpp vs. 0.06 bpp). Notably, the recognition accuracy of videos reconstructed by our method approaches the empirical upper bound set by the original uncompressed sequences, demonstrating that our compression method preserves high-level semantic content with minimal loss.

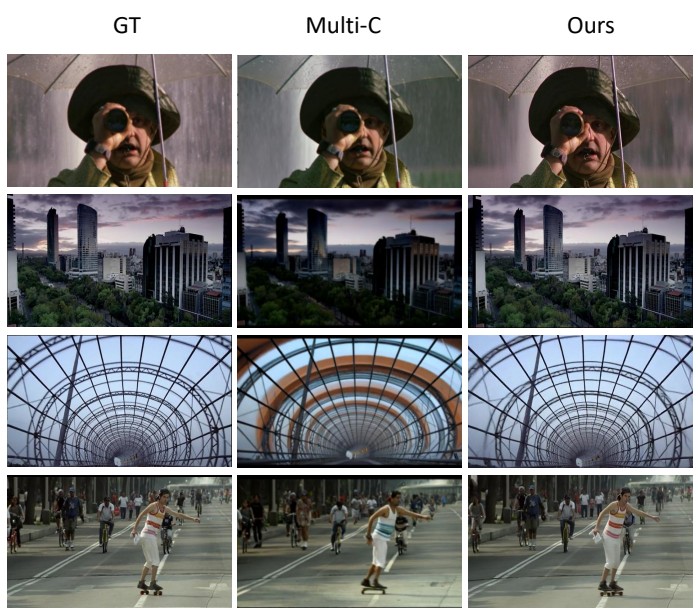

Figure 2: Qualitative comparison of reconstructed video frames under extreme compression on MCL-JCV dataset. Our method (right column) preserves fine details and dynamic elements more faithfully than the hand-crafted feature-based approach by Multi-C (Yi et al., 2025) (middle column), demonstrating the advantage of our data-driven, end-to-end learned representation.

Beyond basic action recognition, we evaluate the codecs on the significantly more challenging multi-object tracking (MOT) benchmark, where identity preservation of human appearance is critical, as well as on the video object segmentation (VOS) task, which requires even more fine-grained semantic features. As shown in Table 2, our approach attains highly competitive performance on all MOT metrics while operating at the lowest bitrate. Although our method falls short of **task-specific** methods fine-tuned for VOS task, it delivers competitive and functionally viable results. This performance gap can be attributed to a fundamental difference in optimization focus: while task-specific methods often employ specialized architectures and losses that explicitly prioritize regions of interest (ROIs) critical for particular tasks, our approach optimizes for holistic visual quality and general semantic fidelity across the entire frame. Figure 3 presents several cases where the reconstructed videos lead to performance degradation in MOT and VOS tasks. While the results exhibit certain blurring and inconsistencies in detailed regions, human observers consistently derive correct semantic information from the content. This suggests that such artifacts, while impactful for automated metrics, remain acceptable in terms of human perceptual tolerance.

In our appendix and supplementary materials, we provide extensive video visualization of reconstructions from the experimental datasets and in-the-wild data. We highly recommend viewing these visual results for a more intuitive and comprehensive understanding of the performance achieved by our method.

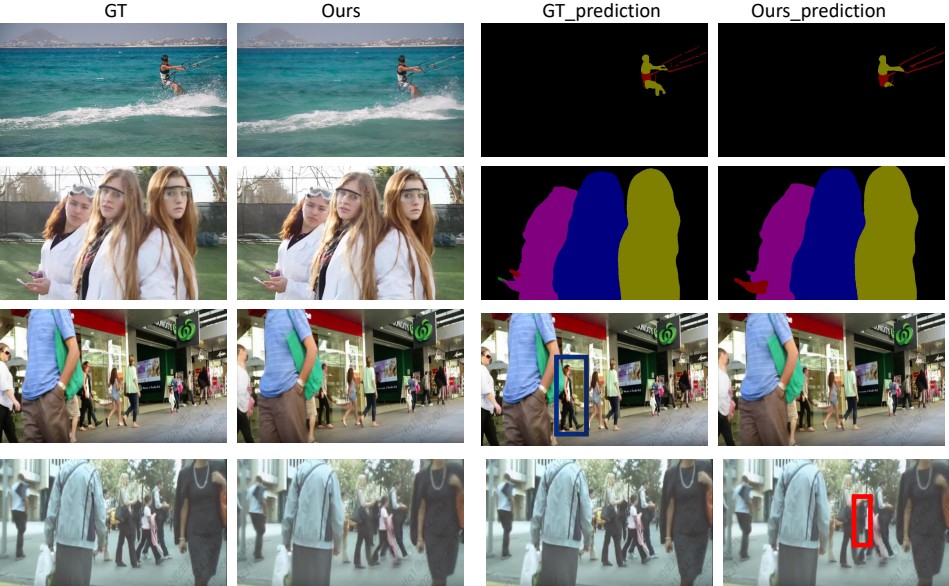

Figure 3: Failure cases on video object segmentation (VOS, first two rows) and multiple object tracking (MOT, last two rows). In the MOT visualization, the blue bounding box denote false negatives, while red bounding box indicate false positives.

## 4.4 ABLATION

To validate the effectiveness of key components in our framework, we conduct extensive ablation studies under consistent experimental settings.

**Effectiveness of the Conditional Prior.** To evaluate the importance of the content-aware conditional prior, we conduct an ablation study in which our proposed prior $p(\mathbf{z}|\mathbf{c})$ is replaced with a fixed standard Gaussian distribution $\mathcal{N}(0, I)$. Experimental results confirm that this change leads to a marked degradation in performance across all key metrics. The KL divergence term, which originally measures only the novel dynamic information absent in the conditioner $\mathbf{c}$, loses its semantic interpretability and fails to effectively constrain the learning of $\mathbf{z}$. As a result, the compressor produces less efficient codes, and the reconstructed video exhibits noticeable artifacts, particularly in regions with complex motion or dynamic textures. These findings empirically validate that our conditional prior plays an essential role in achieving high compression efficiency and maintaining

Table 2: Downstream performances of different coding methods. ↓ denotes the lower is better. 'Upper-bound' is obtained by evaluating the task models with the original videos. The best results are highlighted in **bold**, and the second best results are underlined. For the action recognition task, we report the performance of other methods at a bpp of 0.06; for multiple object tracking (MOT) and video object segmentation (VOS) tasks, we record their performance at a bpp of 0.01. "T-S" denotes "task-specific".

| Method | T-S | Action Rec: TSM on Kinetics | MOT: ByteTrack on MOT17 | | | | VOS: XMEM on DAVIS2017 | | | |
|---|---|---|---|---|---|---|---|---|---|---|
| | | P@1 | MOTA (%)↑ | MOTP (%)↓ | IDF1 (%)↑ | FN↓ | $\mathcal{J}\&\mathcal{F}$ (%) | $\mathcal{J}$ (%) | $\mathcal{F}$ (%) | $\mathcal{F}$-Recall (%) |
| HEVC | ✗ | 35.28 | 61.30 | 19.88 | 64.32 | 17377 | 57.68 | 56.84 | 58.51 | 67.44 |
| FVC | ✗ | 37.23 | 44.24 | 21.97 | 52.53 | 27508 | 62.39 | 61.22 | 63.55 | 75.67 |
| PLVC | ✗ | 48.65 | 67.87 | 18.44 | 68.95 | 13299 | 61.45 | 60.02 | 62.87 | 74.07 |
| DCVC-DC | ✗ | 55.81 | 65.22 | 18.22 | 70.51 | 12328 | 72.86 | 69.26 | 76.46 | 88.17 |
| VVC | ✗ | 49.11 | 64.86 | 19.49 | 68.99 | 15621 | 67.47 | 65.59 | 69.36 | 80.92 |
| ROI | ✓ | 49.82 | 65.29 | 19.31 | 67.78 | 15270 | 69.22 | 67.16 | 71.28 | 83.80 |
| JPD-SE | ✓ | 51.02 | 60.05 | 20.62 | 63.52 | 17847 | 61.48 | 59.86 | 63.09 | 73.31 |
| SMC | ✓ | 59.26 | 70.84 | 17.79 | 71.89 | 11710 | 74.20 | 70.21 | 78.19 | 91.10 |
| FreeVSC | ✓ | 61.83 | **72.80** | **16.61** | **73.21** | 10445 | **76.85** | **72.70** | **80.99** | **92.77** |
| **Ours (bpp ≈ 0.01)** | ✗ | **70.93** | 71.45 | 16.99 | 69.76 | **8594** | 67.31 | 65.61 | 69.01 | 79.85 |
| Upper-bound | N/A | 71.20 | 78.60 | 15.80 | 79.00 | 7000 | 87.70 | 84.06 | 91.33 | 97.02 |

Table 3: Comparison of reconstruction quality between the proposed content-adaptive prior and a standard Gaussian prior. Evaluations are conducted on multiple video quality benchmarks. Our method shows substantial gains across perceptual metrics (e.g., LPIPS, FVD).

| Method | HEVC Class B | | HEVC Class C | | UVG | | MCL-JCV | |
|---|---|---|---|---|---|---|---|---|
| | FVD ($\downarrow$) | LPIPS ($\downarrow$) | FVD ($\downarrow$) | LPIPS ($\downarrow$) | FVD ($\downarrow$) | LPIPS ($\downarrow$) | FVD ($\downarrow$) | LPIPS ($\downarrow$) |
| Baseline | 1092 | 0.3220 | 921 | 0.3025 | 1573 | 0.2931 | 1781 | 0.3568 |
| Ours | **355** | **0.1932** | **311** | **0.1872** | **437** | **0.1645** | **561** | **0.1947** |

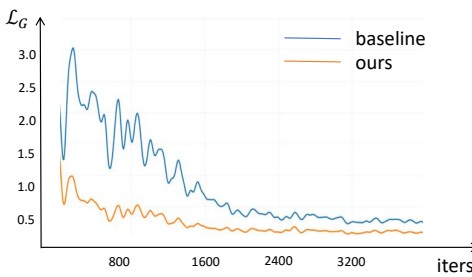

Figure 4: Training loss curves comparing the proposed reformed flow matching objective with the standard noise-conditioned baseline.

Table 4: Comparison of the performance between the noise-to-data baseline and our proposed method under different sampling steps on the UVG dataset.

| Sampling Steps | Method | FVD ($\downarrow$) | LPIPS ($\downarrow$) |
|---|---|---|---|
| 5 | Baseline | 2852 | 0.3556 |
| | Ours | **635** | **0.1938** |
| 10 | Baseline | 2434 | 0.3300 |
| | Ours | **437** | **0.1645** |
| 20 | Baseline | 2003 | 0.3142 |
| | Ours | **415** | **0.1619** |
| 30 | Baseline | 1959 | 0.3074 |
| | Ours | **408** | **0.1615** |

reconstruction fidelity. Table 3 summarizes the results across widely-used video quality assessment datasets. The results demonstrate that our method significantly outperforms the baseline in terms of visual fidelity, with notable improvements in perceptual metrics such as LPIPS and FVD.

**Importance of the Reformed Flow Matching.** To validate the efficacy of our reformed flow matching objective, we compare it against a strong baseline that follows the standard noise-to-data generation paradigm ($\epsilon \rightarrow \mathbf{x}$), where the compressed representation $\mathbf{z}$ is incorporated via channel-wise concatenation with the latent noise, forming the input of the diffusion transformer. The empirical results demonstrate clear advantages of our approach in both the training and inference phases. As shown in Figure 4, our method achieves a faster and more stable convergence, indicating a better optimization behavior. Furthermore, as summarized in Table 4, when evaluating under reduced sampling steps on the UVG dataset, the baseline suffers significant performance decay across metrics. In contrast, our approach maintains consistent high-fidelity reconstruction even in fewer than 10 steps, underscoring its superior efficiency and robustness for low-latency generation.

## 5 CONCLUSION

In conclusion, we have presented a novel generative video compression framework grounded in the Conditional Information Bottleneck principle, which achieves extreme compression ratios (up to 32,768×) while maintaining high reconstruction fidelity and semantic preservation. Our method consistently outperforms both traditional codecs and recent neural alternatives across multiple benchmarks and downstream tasks, demonstrating the effectiveness of integrating generative modeling with information-theoretic learning.

Despite these promising results, several directions remain open for future work. These include further acceleration of the generative decoder via knowledge distillation, model miniaturization for resource-constrained environments, and precise bitrate control for practical adaptive streaming scenarios. We believe this work serves as a strong baseline for semantic-aware video compression and opens up new possibilities for efficient and intelligent visual communication.

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

# A APPENDIX

*This appendix provides more details of our compressor, experimental results, and visualization examples, which are organized as follows:*

• *Architecture of Compressor (cf. § A.1);*

• *Experiment Results (cf. § A.2);*

• *Visualization Example (cf. § A.3).*

## A.1 ARCHITECTURE OF COMPRESSOR

Figure 5 details the network architecture of the Compressor's encoder and decoder. The encoder (left) consists of multiple stages of Attention Layers followed by 3D Convolution Downsampling (Down) modules with specific stride configurations. The downsampling modules employ progressive striding strategies: the first module uses a 1×2×2 stride (maintaining temporal resolution while halving spatial dimensions), followed by two modules with 2×2×2 strides (halving resolution in all temporal and spatial dimensions). The decoder (right) mirrors this structure with 3D Transposed Convolution Upsampling (Up) modules using corresponding stride patterns (2×2×2 and 1×2×2) to progressively restore resolution, interspersed with Attention Layers. The middle section contains Self-Attention and Feed-Forward Network (FFN) modules for feature integration and transformation across the compressed latent space.

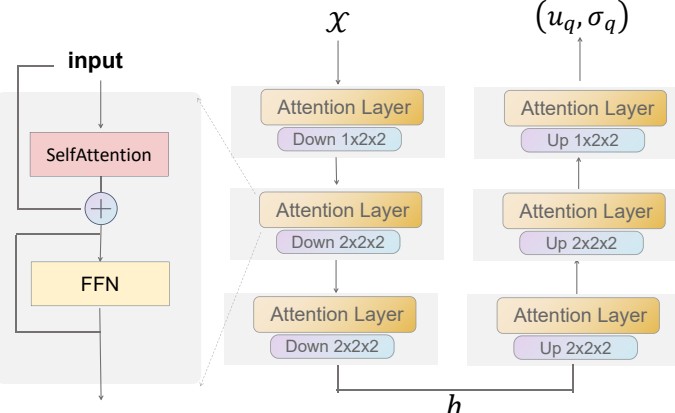

Figure 5: Detailed architecture of the compressor network, featuring a U-Net style design with self-attention layers and spatio-temporal down/up-sampling blocks.

## A.2 EXPERIMENT RESULTS

**Effectiveness of the Conditional Prior.** In the main body, we demonstrate the importance of content-aware conditional prior by benchmarking on multiple standard video compression datasets. To further validate its effectiveness, we also evaluate the reconstructed videos on downstream tasks, including Multi-Object Tracking (MOT) and Video Object Segmentation (VOS). As shown in Table 5, our method consistently outperforms the baseline that uses a standard Gaussian prior, achieving significant improvements across all metrics in both MOT and VOS. These results confirm that the content-adaptive prior not only enhances compression performance but also preserves high-level semantic content more effectively.

**Importance of the Reformed Flow Matching.** Recall that we compare the standard noise-to-data paradigm (which treats the compressed representation as condition and concatenates it along the channel dimension) and our reformed flow-matching objective on the UVG dataset, we now ablate on downstream tasks (e.g., MOT and VOS) to further assess its impact. Table 6. As summarized in Table 6, our method achieves substantial and consistent improvements over the baseline across all evaluation metrics under the same sampling steps, demonstrating that the reformed flow-matching

Table 5: Comparison of downstream task performances between the proposed content-adaptive prior and a standard Gaussian prior. Our method shows consistent gains across multiple tasks.

| Method | MOT | | | | VOS | | | |
|---|---|---|---|---|---|---|---|---|
| | MOTA (%)↑ | MOTP (%)↓ | IDF1 (%)↑ | FN↓ | $\mathcal{J}\&\mathcal{F}$ (%) | $\mathcal{J}$ (%) | $\mathcal{F}$ (%) | $\mathcal{F}$-Recall (%) |
| Baseline | 39.00 | 30.55 | 59.14 | 21826 | 46.31 | 45.33 | 47.28 | 66.57 |
| Ours | **71.45** | **16.99** | **69.76** | **8594** | **67.31** | **65.61** | **69.01** | **79.85** |

objective enables significant performance gains with few sampling steps (i.e., <10), substantially reducing inference overhead.

Table 6: Comparison of downstream task performances between the reformed flow-matching objective and the standard noise-to-data paradigm. Evaluations are conducted on multiple downstream tasks. Our method shows consistent and substantial gains across all the metrics.

| Sampling Steps | Method | MOT | | | | VOS | | | |
|---|---|---|---|---|---|---|---|---|---|
| | | MOTA (%)↑ | MOTP (%)↓ | IDF1 (%)↑ | FN↓ | $\mathcal{J}\&\mathcal{F}$ (%) | $\mathcal{J}$ (%) | $\mathcal{F}$ (%) | $\mathcal{F}$-Recall (%) |
| 10 | Baseline | 27.61 | 32.80 | 52.14 | 26796 | 43.10 | 41.32 | 44.87 | 60.39 |
| | Ours | **71.45** | **16.99** | **69.76** | **8594** | **67.31** | **65.61** | **69.01** | **79.85** |

## A.3 VISUALIZATION EXAMPLE

In this section, we provide additional visual results to further illustrate the model's performance across three aspects: the number of sampling steps, the effect of the conditional prior, and handling of in-the-wild data.

### A.3.1 VISUAL COMPARISON ON DIFFERENT SAMPLE STEPS

To better understand the effectiveness of our method, we provide qualitative visualizations under different settings. As shown in Figure 6, the ground truth, the noise-to-data baseline, and our approach are compared across multiple sampling steps (5, 10, and 20). The baseline tends to produce results with incomplete or blurred structures, especially when the number of steps is small. In contrast, our method consistently generates more coherent and visually faithful patterns, even at fewer steps. With more iterations, the difference becomes increasingly pronounced, where our approach preserves fine-grained details and avoids the artifacts observed in the baseline. These results highlight that our model not only accelerates inference but also improves stability and fidelity in the generated outputs.

### A.3.2 VISUAL COMPARISON ON CONDITIONAL PRIOR

We also compare the visual outcomes of our approach against the baseline where our proposed content-aware conditional prior is replaced with a standard Gaussian distribution. As illustrated in Figure 7, while the baseline produces results that deviate noticeably from the ground truth, our method yields outputs that are structurally more consistent and visually faithful. The baseline often introduces distortions and fails to preserve critical local details, leading to degraded quality. In contrast, our model successfully captures both global structures and fine-grained patterns, resulting in outputs that are closer to the ground truth. These qualitative results provide clear evidence that our approach achieves superior fidelity and robustness compared with the baseline.

### A.3.3 VISUALIZATIONS ON IN-THE-WILD DATA

To further demonstrate the practicality of our approach, we present qualitative results on in-the-wild data, as shown in Figure 8 (see the figure on Page 16). With our novel designs, our method is able to generate outputs that are highly consistent with the ground truth, faithfully preserving both structural integrity and fine details. These results provide strong evidence of the effectiveness of our model beyond the curated datasets. Moreover, the robustness observed across diverse and uncon-

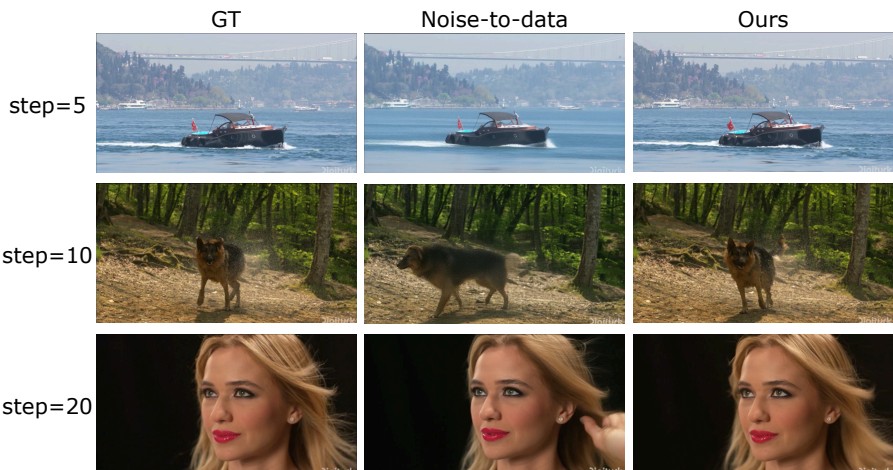

Figure 6: Qualitative comparison across different sampling steps. The baseline (Noise-to-data) suffers from incomplete or blurred structures, particularly at fewer steps, while our method produces more coherent and faithful results with preserved fine details.

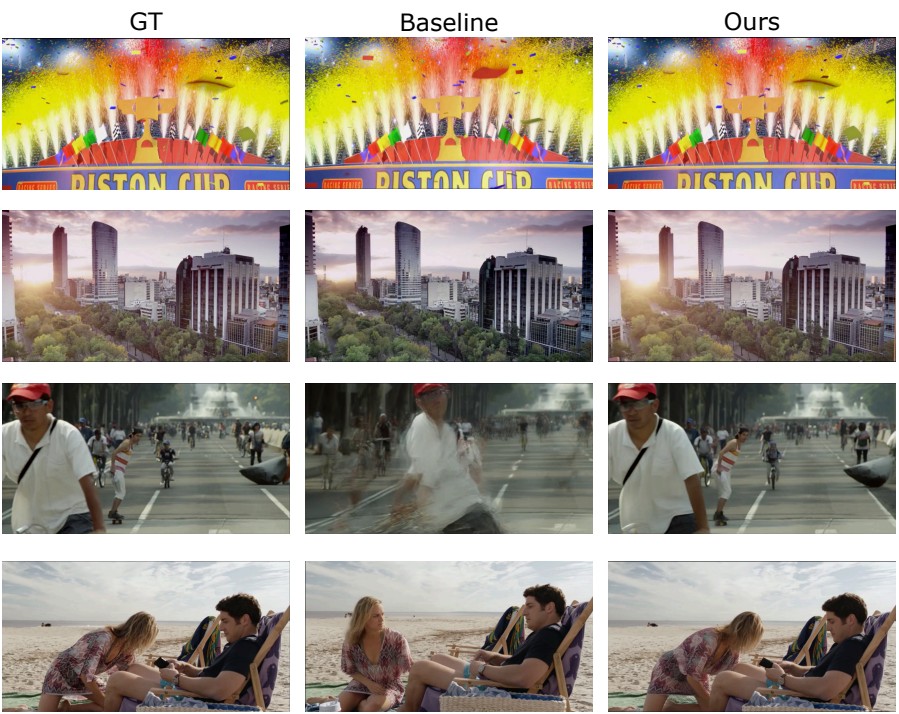

Figure 7: Comparison between the baseline and our approach. The baseline introduces distortions and loses critical local details, whereas our method preserves both global structures and fine-grained patterns, yielding outputs closer to the ground truth.

strained scenarios highlights the generalization capability of our approach, indicating its potential applicability to real-world settings.

For more visualizations, please kindly refer to our supplementary materials, which contains demo videos.

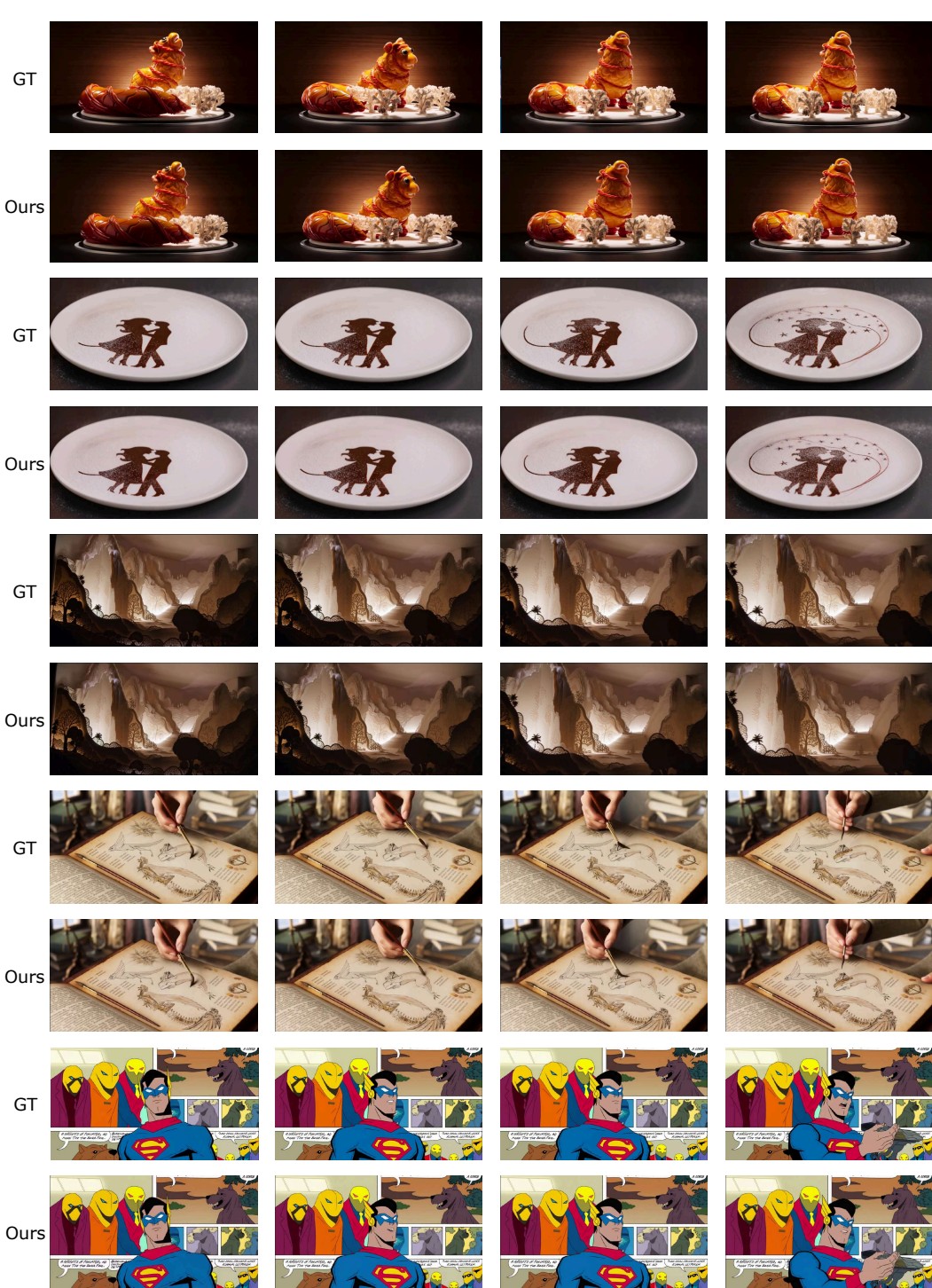

Figure 8: Visualizations on in-the-wild data (best viewed zoomed in), our method faithfully preserves both structural integrity and fine details.

