# OpenReview forum: "Flow-IB: Information Bottleneck Meets Flow Matching for 32,768× Video Compression"
_ICLR.cc/2026/Conference — ICLR 2026 Conference Withdrawn Submission_

### Official Review · Reviewer_yjx3 · 2025-10-30

**Soundness:** 2
**Presentation:** 2
**Contribution:** 3
**Rating:** 2
**Confidence:** 5

**Summary:**

The paper proposes FLOW-IB, a generative video compression framework that transmits only the first and last frames and reconstructs intermediate frames with a flow-matching video diffusion model guided by a conditional information bottleneck objective. The method reports an extreme compression ratio of 32,768× and claims strong perceptual quality (FVD/LPIPS) and competitive downstream task performance at ~0.01 bpp. Key components include (i) a conditional prior derived from temporally masked inputs and a VAE latent space, and (ii) a reformulated flow-matching objective that learns a direct linear path from the compressed latent 𝑧 to the target frames 𝑥, purportedly enabling high-fidelity reconstruction in <10 sampling steps. Experiments compare against H.264/H.265/H.266, DCVC variants, and diffusion-based baselines.

**Strengths:**

1. End-to-end design with IB guidance. The paper integrates a learnable transmitter-side compressor and a receiver-side conditional generator under a variational IB formulation, encouraging the transmitted code to complement (not duplicate) information in the two I-frames. The conditional prior via temporal masking is clearly motivated and computationally tractable in latent space.

2. Reformulated flow-matching objective. Training the generator to follow a linear path from 𝑧 to 𝑥 aligns the generative objective with reconstruction (MSE between 𝑧 and 𝑥) and is argued to reduce sampling steps; the paper provides empirical evidence and ablations versus a noise-to-data baseline.

**Weaknesses:**

1. No full rate–distortion characterization; bpp is approximate. The paper evaluates at (or near) a single bitrate and does not provide RD curves. The reported bitrate is compiled from components and described as “approximately 0.01 bpp,” with Multi-C evaluated at 0.0067 bpp, which complicates a clean, controlled comparison across varying rates. A thorough RD study (multiple operating points with precise bit accounting, including I-frames and side information) is essential for compression claims at this level.

2. Overstated language. The manuscript repeatedly uses terms like “unprecedented” (e.g., “unprecedented 32,768× compression ratio” in the abstract). Such superlatives should be minimized in a technical paper; the results should stand on measured evidence and comprehensive comparisons (including RD curves).

3. Relation to relay/latent-conditioned diffusion is under-analyzed. The paper’s “reformulated flow matching” replaces noise-to-data sampling with a latent-to-data trajectory from 𝑧 (compressed code) to 𝑥. While empirically effective, the work does not provide a rigorous theoretical treatment of equivalence or conditions under which latent-initialized sampling matches (or upper-bounds) noise-initialized diffusion. Prior work such as RDEIC: Accelerating Diffusion-Based Extreme Image Compression with Relay Residual Diffusion (TCSVT’25) explicitly analyzes and motivates relay/latent decoding and discusses its theoretical grounding. Here, beyond the observation that the pathwise objective reduces to an MSE between 𝑧 and 𝑥, the manuscript lacks formal results connecting the proposed objective to maximum-likelihood training of the generative model and to the behavior of standard diffusion samplers. A deeper theoretical comparison (e.g., equivalence, bounds, convergence behavior) to relay diffusion (RDEIC) would strengthen novelty and clarity.

4. Presentation issues. There are several minor but distracting writing/formatting problems (e.g., equation numbering continuity around Eq. (8), typos/notation consistency). For instance, the reparameterization form for 𝑧 is labeled as Eq. (8), but numbering and transitions around this section are untidy, suggesting the need for careful proofreading.

**Questions:**

see the weaknesses.

---

### Official Review · Reviewer_aqyT · 2025-10-30

**Soundness:** 2
**Presentation:** 3
**Contribution:** 2
**Rating:** 2
**Confidence:** 5

**Summary:**

The paper proposes FLOW-IB a generative video compression framework that achieves an extreme claimed ratio of compression (x32,768) by transmitting only the first and last frames plus a tiny latent, and reconstruct all intermediate frames with a flow-matching video generator. The method is derived from a conditional information bottleneck (CIB) objective. Experiments on HEVC Class B/C, UVG, and MCL-JCV report state-of-the-art perceptual metrics (FVD/LPIPS) versus both classical and neural codecs; ablations show the masked prior and reformed flow matching matter. The paper also evaluates semantic fidelity on action recognition (TSM/Kinetics), MOT (ByteTrack/MOT17), and VOS (XMem/DAVIS17), arguing the reconstructions preserve task-relevant content at ultra-low bitrate.

**Strengths:**

- The variational objective based on conditional information bottleneck is clear and principled, and the combination with a content-adaptive prior that explicitly removes boundary-frame redundancy is also intuitive.
- The generative sampling procedure is efficient (under 10 steps) based on the flow matching framework.
- The experiments on (downstream task-relevant) semantic preservations are good and should be considered very important to more general and wider applicability of perceptual/generative video codecs.

**Weaknesses:**

- **Novelty.** I don't think the idea of formalizing video compression with information bottleneck (akin to $\beta$-VAE) is new (refer to [1,2,3] which I think the authors should cite. The idea of masking out/generating all frames but the first and last ones is more of a design choice in my opinion, and the adoption of flow matching for video compression has also been proposed in [4]. So I am not fully agreeing with the statement "first end-to-end video compression framework that integrates generative models with information bottleneck principles".
- **Issue with framework.** The claim of "unprecedented compression ratio of x32,768" is a bit of a overselling because this should not be applicable in real-world applications, simply due to the almost definitely unavoidable loss of distortion/fidelity, drifting, or forgetting caused by only having two references for conditioning. That is, I am challenging the setup and advocating more of a hierarchical bidirectional coding structure or at least more I-/P-frames interleaved in between.
- **Unfair comparison.** First I think the authors should report BD-rate and provide for RD curve comparisons over a wide enough bitrate range, which is more standard in video compression literatures. Further, the comparison against H.265 or DCVC families, which are distortion-oriented, also does not reflect the competence of the proposed model at all due to the totally different focus in reconstruction quality measures. Though I understand that there are much fewer generative video codecs available for comparison, the authors should 1) still report PSNR/MS-SSIM or some other metrics like VMAF which is has a better balance over human perception and content fidelity, and 2) perform more detailed ablations to better demonstrate the superiority of the proposed novelties.
- **Decoding speed.** Decoding speed is particularly important for modern neural video codecs due to how slow they tend to be. Despite the considerable reduction in sampling steps, the base generative model (Wan) should be fairly large and slow, especially on consumer / mobile NPUs/GPUs. The comparison against the baselines would not be fair if the difference in model size is too large, and the decoding speed (on which computing platforms) should also be reported for a more comprehensive evaluation.

Overall, I think the paper is of good quality, well-written, and in general heading towards a right direction of being more perceptual-oriented. However, I do challenge its novelty, and I do not agree with its very aggressive masking strategy, which, might be okey for generative/machine vision tasks, but would most certainly fail in terms of reference-based fidelity checks. Further, there are issues with how the experimental results are presented and compared against benchmark codecs. However, I am will to raise my score based on the authors feedbacks and other reviewers' opinions.

[1] Video Compression With Rate-Distortion Autoencoders, ICCV'19

[2] Variational Lossy Autoencoder, ICLR'17

[3] Deep Generative Video Compression, NeurIPS'19

[4] GIViC: Generative Implicit Video Compression, ICCV'25

**Questions:**

Refer to **Weaknesses**.

---

### Official Review · Reviewer_fs4V · 2025-11-01

**Soundness:** 3
**Presentation:** 2
**Contribution:** 2
**Rating:** 4
**Confidence:** 4

**Summary:**

This work proposes a generative video compression framework that uses the first and last frames as conditions for the reconstruction process. Experimental results show that the proposed approach is comparable to several baselines in reconstruction quality and other downstream tasks.

**Strengths:**

This paper presents a theoretical analysis to derive a loss function based on the provided I-frames.

**Weaknesses:**

- The source of the performance gain in this work is unclear; the improvement in coding performance may stem from the VAE encoder rather than the proposed modules.

- Comparing compression performance at a single rate point does not provide a complete picture of the proposed method’s overall performance.

- Other reconstruction quality metrics, such as PSNR, are not reported.

- Additional codecs that perform well in terms of LPIPS or FVD should be included, such as [1].

[1] Ren Yang, Radu Timofte and Luc Van Gool, "Perceptual Learned Video Compression with Recurrent Conditional GAN", in Processings of the International Joint Conference on Artificial Intelligence (IJCAI), 2022.

**Questions:**

- Since this paper focuses on FVD and LPIPS, it should also compare with other perceptually optimized codecs, such as PLVC, across a wider range of bitrates.

- Focusing solely on FVD and LPIPS does not fully capture reconstruction quality. In the video coding research area, PSNR is generally preferred. Please consider including these additional results in the paper.

---

### Official Review · Reviewer_ibau · 2025-11-10

**Soundness:** 1
**Presentation:** 3
**Contribution:** 2
**Rating:** 2
**Confidence:** 3

**Summary:**

The paper tackles low bitrate video reconstruction with a conditional information bottleneck setup: frames are encoded by a VAE; a latent z is trained with a flow-style objective; training includes a KL term toward a conditional prior p(z|c) computed from a masked version of the video. At test time the system sends a latent plus endpoints, then synthesizes intermediates. Results are reported on standard benchmarks.

**Strengths:**

The overall pipeline is straightforward: transmit a latent and the first and last frames; let the model reconstruct the middle. The method yields competitive rate–distortion given the payload they choose.

**Weaknesses:**

Very confused by the logic behind their implementation of the prior distribution $p(z|c)$. The normal distribution generated by $VAE(\tilde{x})$ just seems like a totally different thing? Like $VAE(\tilde{x})$ is a function of exactly the data you are trying to transmit. It is not a prior that the decoder has access to, and so not a prior you could leverage for compressing/reconstructing a sample from q(z|x). Further, the "compression term" of their objective function (the thing that involves this prior) has no actual bearing on the compression ratio of their final algorithm, since they just transmit the latent Z in bfloat16 form without making use of the prior at all.

re: effectiveness of the conditional prior, the authors try to justify their decision of $p(z|c)$ by showing it works better than a standard normal distribution as the prior. If my interpretation is correct, this changes the compression term of their objective function from being useless to being actively harmful. The ablation I would prefer to see would just be to set the beta coefficient in their objective function to zero and only optimize for reconstruction loss. My hypothesis is that they would end up with the same rate distortion results as before, maybe slightly better.

**Questions:**

Please respond to the weaknesses above. I'd also like to see that beta=0 experiment if you still have time.

---

### Note · Authors · 2025-11-13

**Comment:**

We appreciate the reviewers’ efforts and constructive comments, and we would like to withdraw the submission.

**Withdrawal Confirmation:**

I have read and agree with the venue's withdrawal policy on behalf of myself and my co-authors.